# Transmembrane Protein TMEM230, Regulator of Glial Cell Vascular Mimicry and Endothelial Cell Angiogenesis in High-Grade Heterogeneous Infiltrating Gliomas and Glioblastoma

**DOI:** 10.3390/ijms25073967

**Published:** 2024-04-03

**Authors:** Cinzia Cocola, Edoardo Abeni, Valentina Martino, Eleonora Piscitelli, Paride Pelucchi, Ettore Mosca, Alice Chiodi, Tasnim Mohamed, Mira Palizban, Giovanni Porta, Helga Palizban, Giovanni Nano, Francesco Acquati, Antonino Bruno, Burkhard Greve, Daniela Gerovska, Valerio Magnaghi, Daniela Mazzaccaro, Giovanni Bertalot, James Kehler, Cristiana Balbino, Marcos J. Arauzo-Bravo, Martin Götte, Ileana Zucchi, Rolland A. Reinbold

**Affiliations:** 1Institute of Biomedical Technologies, National Research Council of Italy, 20054 Milan, Italy; cinzia.cocola@itb.cnr.it (C.C.); edoardo.abeni@gmail.com (E.A.); valentinamartino80@gmail.com (V.M.); eleonora.piscitelli@itb.cnr.it (E.P.); paride.pelucchi@itb.cnr.it (P.P.); ettore.mosca@itb.cnr.it (E.M.); alice.chiodi@itb.cnr.it (A.C.); ileana.zucchi@itb.cnr.it (I.Z.); 2Department of Pharmacological and Biomolecular Sciences, University of Milan, 20133 Milan, Italy; tasnim.mohamed@unimi.it (T.M.); valerio.magnaghi@unimi.it (V.M.); 3Department of Gynecology and Obstetrics, University Hospital of Münster, 48149 Münster, Germany; mira.palizban@uni-muenster.de (M.P.); drhelgapalizban@me.com (H.P.); martin.goette@ukmuenster.de (M.G.); 4Center for Genomic Medicine, Department of Medicine and Surgery, University of Insubria, 21100 Varese, Italy; giovanni.porta@uninsubria.it; 5Operative Unit of Vascular Surgery, I.R.C.C.S. Policlinico San Donato, 20097 San Donato Milanese, Italy; giovanni.nano@unimi.it (G.N.); daniela.mazzaccaro@gmail.com (D.M.); 6Department of Biomedical Sciences for Health, University of Milan, 20122 Milan, Italy; 7Human Genetics Laboratory, Department of Biotechnology and Life Sciences, University of Insubria, 21100 Varese, Italy; francesco.acquati@uninsubria.it; 8Laboratory of Immunology and General Pathology, Department of Biotechnologies and Life Sciences, University of Insubria, 21100 Varese, Italy; 82antonino.bruno@gmail.com; 9Laboratory of Innate Immunity, Unit of Molecular Pathology, Biochemistry, and Immunology, I.R.C.C.S. MultiMedica, 20138 Milan, Italy; 10Department of Radiation Therapy and Radiation Oncology, University Hospital of Münster, 48149 Münster, Germany; greveb@uni-muenster.de; 11Computational Biology and Systems Biomedicine, Biogipuzkoa Health Research Institute, Calle Doctor Begiristain s/n, 20014 San Sebastian, Spain; daniela.gerovska@biodonostia.org (D.G.); marcos.arauzo@biodonostia.org (M.J.A.-B.); 12Department of Anatomy and Pathological Histology, Santa Chiara Hospital, APSS, 31822 Trento, Italy; giovanni.bertalot@apss.tn.it; 13Centre for Medical Sciences—CISMed, University of Trento, 38122 Trento, Italy; 14Laboratory of Cell and Molecular Biology, NIDDK, National Institutes of Health, Bethesda, MD 20892, USA; james.kehler@icloud.com; 15I.R.C.C.S. Ospedale Galeazzi-Sant Ambrogio, 20157 Milan, Italy; balbino.cristiana@yahoo.it; 16Basque Foundation for Science, IKERBASQUE, Calle María Díaz Harokoa 3, 48013 Bilbao, Spain; 17Department of Cell Biology and Histology, Faculty of Medicine and Nursing, University of Basque Country (UPV/EHU), 48940 Leioa, Spain

**Keywords:** angiogenesis, vascular mimicry, metalloproteinases, vesicles, endoplasmic reticulum, TMEM230 (C20orf30), Golgi complex, U87-MG, glycoproteins, endothelial cells, high-grade diffuse infiltrating gliomas, astrocytoma, microchannels, tumor microtubules, heparinase, heparan sulfate proteoglycans, RNASET2

## Abstract

High-grade gliomas (HGGs) and glioblastoma multiforme (GBM) are characterized by a heterogeneous and aggressive population of tissue-infiltrating cells that promote both destructive tissue remodeling and aberrant vascularization of the brain. The formation of defective and permeable blood vessels and microchannels and destructive tissue remodeling prevent efficient vascular delivery of pharmacological agents to tumor cells and are the significant reason why therapeutic chemotherapy and immunotherapy intervention are primarily ineffective. Vessel-forming endothelial cells and microchannel-forming glial cells that recapitulate vascular mimicry have both infiltration and destructive remodeling tissue capacities. The transmembrane protein TMEM230 (C20orf30) is a master regulator of infiltration, sprouting of endothelial cells, and microchannel formation of glial and phagocytic cells. A high level of TMEM230 expression was identified in patients with HGG, GBM, and U87-MG cells. In this study, we identified candidate genes and molecular pathways that support that aberrantly elevated levels of TMEM230 play an important role in regulating genes associated with the initial stages of cell infiltration and blood vessel and microchannel (also referred to as tumor microtubule) formation in the progression from low-grade to high-grade gliomas. As TMEM230 regulates infiltration, vascularization, and tissue destruction capacities of diverse cell types in the brain, TMEM230 is a promising cancer target for heterogeneous HGG tumors.

## 1. Introduction

Glial cells provide physical and chemical support for the homeostasis of the extracellular compartment of neural tissue through direct contacts and the secretion of soluble factors, vesicles, and insoluble and substrate-forming scaffolds [1]. Following injury, glial cells of the central nervous system promote tissue remodeling by microchanneling, new blood vessel formation, and scar formation, processes associated with normal wound healing [2,3,4]. Microchannel lumen structures, besides promoting passive permeability and the circulation of oxygen and nutrients, also allow for the diffusion of wound healing and angiogenesis-promoting factors (Figure 1). In wound healing, in addition to glial cells, phagocytic cells, such as monocytes and macrophages, also promote microchanneling. Likely, microchannels also provide a “path” that allow sprouting cells, such as endothelial cells, to migrate and form new blood vessels.

In neural disease or tumor development, the aberrant expression and secretion of glial and phagocytic cell factors result in destructive tissue remodeling and loss of normal blood vessel structure, formation, and function. Microchannels also promote destructive tissue remodeling by forming lumen and scars. The destruction of blood vessels and the formation of microchannels and scar tissue promote the loss of normal vascular activities and drug delivery for cancer treatment. We previously demonstrated that the transmembrane protein TMEM230 regulates both tissue vascularization by inducing endothelial cell sprouting and vessel formation and vascular mimicry by regulating glial- and macrophage-cell-generating microchannels [5]. TMEM230 is an evolutionarily conserved multifunctional protein expressed in various cell types such as tubule-forming glandular cells and endothelial cells [5,6]. TMEM230 has both intracellular and extracellular trafficking and secretion activities. For instance, secreted factors and vesicles from U87-MG cells (a glial cell line model of glioblastoma multiforme, GBM) expressing TMEM230 have the capacity to promote endothelial cell sprouting and proliferation. TMEM230 promotes the intracellular and extracellular trafficking of the signaling factors of angiogenesis by regulating endothelial tip- and stalk-cell formation [6]. Sustained over-expression of TMEM230 in endothelial cells also promoted loss of normal blood vessel structure and function by inducing loss of cell-to-cell contacts [6]. Recently, we demonstrated that TMEM230 induces vascular mimicry by secretion of glial and macrophage cellular proteins and glycan-digesting enzymes and glycoproteins that have microchannel- and scar-forming capacity [5]. The addition of secreted factors generated from glial cells in which TMEM230 expression was upregulated to human umbilical vein endothelial cell (HUVEC) cultures resulted in endothelial cell sprouting and blood-vessel-like structure formation. Collectively, our previous studies have supported that expression of TMEM230 promotes the infiltration of various cell types [5], including microglia, macrophages, endothelial cells, and immune cells.

TMEM230 regulates the trafficking of blood-vessel-forming signaling factors in endothelial cells and the extracellular secretion of tissue-digesting enzymes in microchannel-forming (mimicking vasculogenesis) glial and phagocytic cells. We therefore hypothesized that the destructive tissue remodeling associated with HGG and GBM was due to the contribution of various cell types, including endothelial, glial, and macrophage cells that express elevated levels of TMEM230. High-grade gliomas and glioblastoma are highly aggressive tumors and appear heterogeneous cellularly and functionally. This heterogeneity contributes to variability in destructive neural tissue remodeling, aberrant blood vessel formation, and response to conventional and state-of-the-art anti-cancer therapies in patients [7]. 

As HGG and GBM are associated with highly infiltrating cells, we propose that HGG and GBM cell heterogeneity may be due to aberrant overexpression of TMEM230 in various cell types with infiltration and vessel and microchannel formation capacity.

Progress in glioma research would greatly benefit by understanding which genes, molecular pathways, and diverse cell types are regulated by TMEM230 and contribute to tissue infiltration, de novo formation of blood vessels, vascular mimicry, and microchannel formation in highly vascularized tumors such as glioblastoma multiforme and high-grade oligodendroglioma (ODG). Research also supports that low-grade oligodendrogliomas gradually become more aggressive over time and progress to high-grade gliomas [8,9,10]. If this tumor progression model is valid, and aberrant elevated levels of TMEM230 promote aggressive tumor development by modulating the infiltrating properties of diverse types, TMEM230 may represent a significant target for cancer therapeutic research. 

GBM and HGG are the most aggressive tumors originating in the brain, with histopathologic features that include disorganized and highly permeable blood vessels and extensive cellular processes/projections and cells that infiltrate into the brain parenchyma and tissue [11,12,13,14]. These invasive cells and their processes are also associated with contact and the extensive remodeling capacity of existing blood vessels. As previously mentioned, the infiltration of cells and their cellular projections also allow new blood vessels to form by providing microchannels for sprouting endothelial cells to migrate. Known targets for anti-angiogenic therapies provide minimal or no effect in the overall survival of 12 to 15 months following diagnosis in patients with GBM [7,15,16,17]. A likely cause of the inability to treat GMB is that aberrantly formed blood vessels and extensive microchannel formation contribute to highly permeable “leaky” vasculature of the brain, which results in intermittent blood flow and the inability of the cardiovascular system to effectively direct therapeutic agents to tumor cells. Limited research into the role of vascular mimicry in HGG/GMB development greatly inhibits the identification of novel genes and molecular pathways that promote destructive tissue remodeling by microchannel-forming tissue-digesting enzymes and factors that regulate tissue digesting enzymes. 

Identifying novel targets for tumor-induced angiogenesis and vascular mimicry therefore remains an important goal for developing effective treatments for highly vascularized tumors such as GBM or high-grade ODG generated by diverse infiltrating cells. Characterizing the interactions of endothelial, glial, and macrophage cells in HGG cells may provide a deeper understanding of why it is difficult to identify anti-cancer agents for treating tumors with heterogeneous cell types. We previously identified TMEM230 as a master regulator of the sprouting of endothelial cells in vertebrate early development [5]. Additionally, we demonstrated that TMEM230 promotes vascular mimicry, a process generated by different cell types with microchannel formation capacity [2,5]. Microchannels formed during tissue wound healing are different from blood vessels formed in angiogenesis (Figure 1). Microchannels differ with bona fide blood vessels in that they are not surrounded by a “wall” of cells but are lumen structures that are supported instead by the scaffolds or matrix of the tissue [18]. Therefore, the lumen permeability of microchannels is not regulated at the cellular level, allowing the unregulated diffusion and permeability of diverse factors into the tissue mass.

## 2. Results

### 2.1. Transmembrane Protein TMEM230 Expression Is Necessary for Endothelial and Glial Cell Adhesion, Sprouting, Migration, and Infiltration

Our research originally identified TMEM230 as essential to maintain normal blood function through its role in regulating endothelial cell-to-cell substratum adhesion and therefore the structural integrity of blood vessels and blood vessel impermeability. These activities were essential to promote proper blood vessel network formation and blood vessel renormalization in the early development of zebrafish. Ablation of the expression TMEM230 in zebrafish showed that dorsal endothelial cells lost the ability to sprout, migrate, and maintain normal cell-to-cell contacts, resulting in vascular insufficiency in developing embryos. Aberrantly high levels of TMEM230 resulted in the hypervascularization of embryos due to the generation of permeable and highly invasive blood vessels and disorganized infiltration of endothelial cells. These characteristic features of highly vascularized tissue and defective blood vessels are associated with aggressive gliomas such as glioblastoma multiforme and high-grade oligodendroglioma. An aggressive tumor property also observed in GBM and HGG-ODG is the destructive remodeling of neural tissue and neural parenchyma [19]. Oligodendrogliomas are associated with infiltrating oligodendrocytes or glial precursor cells and may, like other types of infiltrating gliomas, such as GBM, invade around the endothelial cells of the blood vessels of the brain. This results in the displacement, remodeling, or destruction of blood vessels. The infiltration and displacement of endothelial cells was previously observed by our group to be regulated by TMEM230 in U87-MG cells (Figure 2).

Like the behavior of TMEM230 in HUVECs, the expression of TMEM230 in 2D and 3D cell cultures was found necessary to maintain the cell morphology, viability, and microchanneling of U87-MG glial cells (Figure 2 and Figure 3). TMEM230 expression was shown to be necessary for U87-MG substratum attachment and survival capacity by the observation that when TMEM230 was downregulated (shTMEM230+eGFP), U87-MG lost normal cell morphology and maintenance of cellular cytoplasmic-like invadopodium and projections and detached in 2D adherent cultures (Figure 3). The role of TMEM230 in U87-MG cells was similarly observed in endothelial cells in in vitro and in vivo assays, in which the ablation of TMEM230 promoted loss of normal cell morphology and survival.

### 2.2. Endogenous Expression of TMEM230 Promotes “Tunneling”, Microchannel Formation, Extracellular Matrix Infiltration, and Vascular Mimicry

The loss of the structural morphology of glial cells and the extracellular adhesion of cells to the substratum was likely due to the inability to maintain the trafficking and renewal of intracellular scaffolds and the secretion of scaffold components onto substratum, a function regulated by the Golgi apparatus. When grown in Matrigel culture conditions, U87-MG cells in which TMEM230 was downregulated (shTMEM230+eGFP), compared to control cells (shSCR+eGFP), lost the ability to migrate and form microchannels and lumen, supporting the essential role of scaffolds in cell morphology, movement, and the secretion of extracellular matrix and scaffold-digesting enzymes (Figure 4, bright field image panels on the right). This behavior was originally observed in HUVECs when TMEM230 was downregulated or when HUVECs were cultured in Matrigel with conditioned media obtained from 3-day cultures of U87 control (shSCR+eGFP) and U87 in which TMEM230 was downregulated (shTMEM230+eGFP) (Figure 4, 2 panels on the right).

### 2.3. High Levels of Transmembrane Protein TMEM230 Are Associated with Lower Survivability in Patients with High-Grade Oligodendroglioma

Research supports that gliomas such as low-grade ODG gradually become more aggressive over time [20]. Aggressiveness may be due to the destructive remodeling of tissue by infiltrating glial cells and the generation of aberrantly formed defective blood vessels and microchanneling, leading to the inability to deliver and target tumor cells in neural tissue [21]. To evaluate whether the tumor grade progression model (i.e., progression from LGG to HGG) was associated with aberrantly elevated levels of TMEM230 expression, open-access mRNA sequencing datasets were analyzed from patients with low-grade glioma (LGG), patients with high-grade glioma (HGG), and patients with glioblastoma multiforme (GBM). Approximately 200 patients with ODG and a cohort of 172 patient samples with GBM from The Cancer Genome Atlas (TCGA) RNA sequencing (RNAseq) database were analyzed for high and low TMEM230 expression levels (https://www.cancer.gov/ccg/research/genome-sequencing/tcga) accessed on 1 December 2023. Analyses were performed using the TCGA2STAT R Package, as previously described [5]. Expression data were used to determine whether TMEM230 was also differentially expressed in glial tumor tissue cells from patients with low- or high-grade ODG or GBM (Figure 5 and Figure 6 and Appendix A). Appendix A lists genes differentially expressed in patients with LLG and HGG oligodendroglioma and patients with GBM (high-grade) correlated with high and low TMEM230 expression.

Patient-derived tumor gene expression analyses supported that TMEM230 has prognostic value as a tumor marker for aggressive HGG-ODG since a higher level of TMEM230 was associated with lower patient survival (Figure 5 and Figure 6) and worse prognosis. A higher percentage of patients died more rapidly compared to patients with lower expression of TMEM230. High expression of TMEM230 was therefore associated with low survivability for patients with HGG-ODG (Figure 6). Patients with GBM expressed the highest levels of TMEM230 (Figure 5) and the lowest survivability (Figure 6) when compared to patients with LGG or HGG oligodendroglioma. Even the lowest levels of TMEM230 in GBM corresponded to the high levels of HGG oligodendrogliomas. As the TMEM230 expression of HGG oligodendroglioma corresponded with low survivability for patients, not surprisingly, low survival was also correlated with GBM. Therefore, no correlation could be generated with patient survival using differential expression of TMEM230 in GBM (*p*-value 0.8494, Figure 6) as almost all patients with GBM expressed very high levels of TMEM230 and almost all patients had low survival after 4 years (1500 days). This supports that the low levels of TMEM230 in patients with GBM were not low enough to be protective against high patient mortality.

Collectively, patient-derived tumor expression analysis supported that a high level of TMEM230 was associated with high-grade infiltrating and more aggressive gliomas (in terms of patient survival and tumor grade). As elevated TMEM230 expression was a prognostic marker for HGG, we investigated genes and pathways that were differentially expressed in patient tumors when TMEM230 was also differentially expressed to determine why high levels of TMEM230 are not protective against, or why high TMEM230 may promote, low survivability.

Of interest was to determine which genes and pathways may be promoted by elevated levels of TMEM230 in the progression from low- to high-grade gliomas and whether these genes and pathways were associated with angiogenesis or microchannel formation.

Figure 5 and Figure 6 support that in GBM no correlation exists between patient survival and high and low expression of TMEM230. As GBM has predominantly higher levels of TMEM230 compared to both LGG and HGG oligodendroglioma, this suggests that even low TMEM230 expression in GBM is sufficiently high to induce high patient mortality.

### 2.4. Candidate Pathways Regulated by TMEM230 in HGG and GBM

The cell assays in Figure 4 supported that TMEM230 had an essential role in microchannel formation by regulating the intracellular trafficking and secretion of scaffold-digesting enzymes, such as metalloproteinases. The endoplasmic reticulum and Golgi complex are the hub of endomembrane trafficking and secretion, powered by motor proteins [22,23,24]. In addition to microchannel formation, the intracellular trafficking and secretion of scaffold-digesting enzymes, such as metalloproteinases, are also essential in angiogenesis. To evaluate whether TMEM230 is a regulator of the endomembrane system, candidate genes and pathways regulated or co-regulated with TMEM230 expression in ODG and GBM tumors were analyzed (Appendix A and Figure 7). All genes that had significant differential expression with a *p*-value adjusted to ≤0.05 when TMEM230 was also differentially expressed (*p*-value adjusted to ≤0.05) were analyzed (Appendix A). Gene ontology and biological pathways were then assessed for the genes differentially expressed using the False Discovery Rate method.

In support of the functional cellular analysis (Figure 4), high expression of TMEM230 was significantly associated with genes regulating endomembrane end-product synthesis or trafficking in the organelles, endoplasmic reticulum (ER), and Golgi apparatus as indicated by predominant upregulation of proteoglycans and glycosylation genes (Appendix A). N-glycosylation is a process that occurs in the endoplasmic reticulum (ER) and Golgi body. Initial synthesis of precursor molecules occurs in the ER, with subsequent processing occurring in the Golgi complex. The expression of these organelle genes and genes of the endocytic vesicle membrane supports that TMEM230 regulates endomembrane trafficking and secretion.

In support that TMEM230 regulates the endomembrane system, genes associated with factors for vesicle trafficking and secretion were identified (Appendix A). The upregulation of metalloproteinases (such as A Disintegrin and Metalloproteinase (ADAMs) and Matrix Metalloproteinases (MMPs), in bold in Appendix A) and phagosome genes supports that TMEM230 directly regulates microchannel formation by secretion of scaffold-digesting enzymes. The trafficking and secretion of these enzymes are powered by motor proteins, essential in the movement of the intracellular and extracellular trafficking of cargo (Appendix A). Phagosomes are vesicles of the endomembrane system formed around the material that enters a cell by phagocytosis.

Collectively, the pathways and genes uncovered supported that TMEM230 has a role in the shuttling and secretion of microchannel-forming metalloproteinases, ADAMs and MMPs, and phagosomes. The Golgi complex, combined with the endoplasmic reticulum, is the hub of all cargo intracellular and extracellular trafficking and secretion (Figure 7) [22,23,24]. The motor-protein-dependent cargo trafficking of intracellular and extracellular factors, membrane components, and vesicles is dependent on their physical interactions with cytoskeletal scaffolds. In turn, cytoskeletal scaffold renewal and maintenance are dependent on endomembrane trafficking. In addition to regulating secretion for microchannel formation, TMEM230 is also likely essential in maintaining or modulating cell polarity, 3D tissue and cell structure and function, cell-to-cell contacts, cell-to-substrate adhesion, and cell motility, as observed in Figure 3 [25,26]. Additionally, cargo trafficking is essential in motor-protein-dependent cytoskeletal remodeling for generating extracellular projections and processes for cell sprouting and infiltration into tissue (Figure 3 and Figure 7) [25,26,27,28].

Cell functional assays using U87-MG cells support that TMEM230 induced microchannel formation by upregulating the endosome system of glial cells, specifically Golgi complex activity, indicated by the upregulation of proteoglycans and glycosylation activity. Microchannel formation is a biological process that recapitulates vascular mimicry and angiogenesis by glial or macrophage cells. Similarly, endogenous TMEM230 expression was shown to induce cell sprouting, migration, and blood vessel formation in HUVECs (Figure 4). As terminal glycosylation of glycoproteins occurs in the Golgi apparatus and is necessary for endothelial sprouting and tissue infiltration, we investigated whether TMEM230 also regulates glycoprotein expression in infiltrating ODG. High levels of TMEM230 in ODG were found to be associated with the upregulation of glycoproteins and angiogenesis-associated genes (Appendix A and Figure 8 and Figure 9). This suggested that high expression of TMEM230 promoted glycoprotein-associated angiogenesis in highly vascularized infiltrating gliomas.

As a very large number of genes (734 glycoprotein genes and 46 genes associated with angiogenesis (Appendix A)) were found to be modulated with the upregulation of TMEM230 expression in ODG, to identify the most significantly modulated glycoproteins and genes in angiogenesis, genes were selected based on a more stringent adjusted *p*-value (<1 × 10^−5^) and an absolute log2 fold change of >2, as shown in Appendix A.

As noted previously, patients with GBM expressed the highest levels of TMEM230 (Figure 5) and were associated with lowest survivability (Figure 6) when compared to patients with ODG. No correlation was observed between patient survival and high and low expression of TMEM230 in GBM (Kaplan–Meier, *p*-value 0.8494; Figure 6). Similarly, most glycoproteins and genes associated with angiogenesis displayed no fold change in expression or were correlated with a fold change associated with an adjusted *p*-value with no significance, suggesting that TMEM230 had lost the ability to regulate these genes in GBM (Appendix A).

We hypothesize that any level of TMEM230 expressed in GBM constitutively maintained these genes in an elevated state of expression compared to ODG. This is supported by the observation that glycoproteins and genes associated with angiogenesis were expressed at higher levels in patients with GBM compared to ODG (see base mean expression levels, Appendix A). The base mean expression levels of these genes were 10 to 60 times higher in GBM with respect to LGG, supporting that even the low levels of TMEM230 in GBM maintains elevated levels of Golgi complex (indicated by glycoprotein expression) and angiogenesis activities.

Our results supported that endomembrane and secretion activities were driven by TMEM230 and that these activities were aberrantly elevated in GBM and ODG in which TMEM230 was upregulated and were associated with lower survivability. We hypothesized that elevated levels of TMEM230 drive both glioma (or macrophage) cell infiltration through microchannel formation and endothelial cell sprouting for blood vessel formation (Figure 8 and Figure 9). High-grade gliomas (ODG or GBM) are characterized by heterogenous and aggressive populations of tissue-infiltrating cells that promote both destructive tissue remodeling and aberrant vascularization of the brain. Our results may support that TMEM230 likely regulates various cell types in heterogeneous infiltrating gliomas by modulating microchannel-forming glial cells and blood-vessel-forming endothelial cells and these activities likely overlap. That is, microchannels may directly interact and promote aberrant blood vessel formation by allowing the diffusion of proangiogenic factors (Figure 9). In support of this, our research showed that microchannel-forming glial cells, physically contact endothelial cells and promote altered behavior in endothelial cells and blood vessels (see Figure 2).

In addition, the formation of defective and permeable blood vessels and microchannels and destructive tissue remodeling may prevent the vascular delivery of pharmacological agents to tumor cells and represent the main reason why therapeutic chemotherapy and immunotherapy intervention are ineffective.

### 2.5. Astrocytoma Patient Data Set Analysis

Our hypothesis was that high-grade glioma behavior was due to the aggressive infiltration of brain tissue associated with blood-vessel- and microchannel-forming cells. Determining whether other glioma cell types, such as astrocytoma, have similar pathways modulated with TMEM230 expression levels would provide further evidence that TMEM230 may be a cancer target for glioma treatment. We performed similar statistical analyses using the transcriptomic data of astrocytoma patients. Astrocytomas are the second most common gliomas and are associated with extreme infiltration, making surgical removal nearly impossible. Prognosis for grade 4 astrocytoma (grade IV GBM) is 3 years or less [8,9,10]. TMEM230 was significantly differentially expressed between LGG astrocytoma compared to HGG astrocytoma (a base mean expression of 1872.54 in LGG tumors, and a log2 fold change of 0.695271080072909 associated with an adjusted *p*-value of 9.14930075327835 × 10^−74^). The genes and pathways identified regulating infiltration and microchanneling that were modulated with TMEM230 expression were similarly identified in astrocytoma as they were for ODG (Appendix A).

The genes and pathways identified as being regulated by TMEM230 were supported by the astrocytoma cell functional assays (Figure 10). As for U87-MG cells, infiltration and microchanneling were inhibited with the ablation of TMEM230 in astrocytoma cells (Figure 10).

Collectively, the results of the infiltrating astrocytoma cells support that TMEM230 is regulator of glial cell vascular mimicry and endothelial cell angiogenesis in diverse high-grade infiltrating gliomas, including ODG and astrocytoma.

### 2.6. VEGF Analysis

TMEM230 activity in endothelial tip-cell sprouting was previously shown to be independent from the VEGF signaling pathway in an animal model [6]. The independence of activity was supported in zebrafish by TMEM230 being able to promote angiogenesis with impairment of the VEGF activity and by the downregulation of TMEM230 inhibiting angiogenesis with upregulation of VEGF signaling [6]. During angiogenesis, new vessels are generated from existing endothelial cells that become tip cells that promote degradation of the vascular extracellular matrix, a process recapitulating microchannel formation through the secretion of metalloproteinases. Appendix A supports that TMEM230 is associated with angiogenesis in gliomas. To determine whether TMEM230 is a master regulator and independent of the VEGF signaling pathway in glioma formation and angiogenesis (Figure 4 and Appendix A), HUVEC cultures were treated with VEGF ligand (20 ng/mL), and TMEM230 expression was modulated (Figure 11). The cell assays show that, as expected, VEGF promoted sprouting (panels 1, 3) compared to cultures where VEGF was not added (panels 2, 4). Transgenic expression of TMEM230 mRNA also promoted sprouting (panels 5, 6), regardless of whether VEGF was added or not to the cultures. This suggests that TMEM230 can promote angiogenesis and microchanneling independently of VEGF ligand activity (panel 6). Whether TMEM230 downregulation can impair VEGF-promoted cell activities was indicated by the inhibition of cell sprouting and infiltration in cultures in which VEGF was present and in which TMEM230 was downregulated (see panels 7 and 8). This strongly supports that TMEM230 downregulation may represent an effective anti-cancer therapy in glioma treatment. Further insight was obtained when the number of sprouts and their respective lengths were quantified (Figure 12). Lane 5, in Figure 12, may suggest that TMEM230 and VEGF may act synergistically, where the sproutings are not significantly different but their numbers have increased.

### 2.7. Hypoxia Analysis

Of interest and clinical importance is whether TMEM230 regulates tissue response to hypoxia and may be upregulated in the hypoxic tumor microenvironment. Our previous study suggested that GBM pathophysiology was characterized by aberrant hypervascularization in which defective and highly circuitous permeable blood vessels are formed [5]. This creates a tumor tissue microenvironment that is hypoxic, resulting in both tissue necrosis and cell death, in agreement with the lower patient survival associated with higher levels of TMEM230 in oligodendroglioma and GBM (Figure 5 and Figure 6). Aggressive high-grade gliomas are highly vascularized with permeable and defective blood vessels that result in tissue hypoxia, necrosis, and cell death, suggesting that TMEM230 may be upregulated in part due to lack of oxygen. Appendix A shows that increase in TMEM230 expression was correlated with increased expression of genes associated with hypoxia in oligodendroglioma and astrocytoma.

Whether TMEM230 expression was upregulated by hypoxia signaling or TMEM230 elevated expression-induced upregulation of hypoxia genes was analyzed by culturing HUVECs in normoxia or hypoxia (1%) (Figure 13). Upregulation of TMEM230 was observed in cells cultured in hypoxia (lane 2 compared to lane 1) or with cells treated with VEGF ligand in normoxia (lane 3 compared to lane 1). The capacity for hypoxia to upregulate TMEM230 apparently was less than the capacity for VEGF ligand (lane 3 compared to lane 4 and lane 2 compared to lane 3).

## 3. Discussion

Glial cells provide physical and chemical support and protection for neurons, neural tissue, and diverse cell types of the brain. Secreted factors, scaffolds, and vesicles regulate, in addition to the normal homeostasis of the brain, the tumor micro-environment of GBM and HGG. GBM and HGG are highly aggressive tumors with bad prognosis and poor response to all cancer treatments [9,12]. They are characterized by high heterogeneity at the cellular level and high infiltration. Infiltration is a property of various cell types of the CNS, including immune cells, macrophages, and endothelial cells [13,14]. Infiltrating tumor cells are also commonly associated with aberrant and high vascularization, where infiltration and vascularization promote destructive tissue remodeling and loss of normal vascular function and, consequently, the inability to deliver therapeutic agents to tumor cells. Aberrantly secreted factors and vesicles have diverse roles in the pathological formation of blood vessels and in 3D destructive tissue remodeling [30]. In pathological conditions, glial cells secrete factors, vesicles, and scaffolds that promote extracellular matrix digestion, resulting in microchannel formation and scar formation [31,32]. These microchannels have various functions depending on the anatomical location of the glial cells. In the CNS, localized glial cells generate microchannels that recapitulate vascular mimicry and wound healing processes. Microchannels are dynamic structures that promote (1) digestion of polymeric macromolecules of tissue, (2) absorption and circulation of tissue-soluble factors, and (3) deposition of newly synthesized scaffolds into lumen (Figure 9). In contrast to bona fide vessels, such as blood vessels, microchannels are structures in which lumen is not surrounded by cells, and therefore lumen permeability is not regulated at the cellular level. Deposition of fibrous scaffolds into luminal space provides traction for glial or neural cell migration and cytoplasmic extensions such as cellular processes and axons. Extensions also allow the formation of new connections and endothelial cell migration during blood vessel formation. GBM is associated with a heterogeneous population of phagocytic tumor cells, predominantly glial and macrophages. Specific factors secreted by phagocytic cells were identified with tissue and blood vessel remodeling capacities (Appendix A). These include plasma membrane, basement membrane and tissue extracellular scaffold degrading enzymes such as heparanase that cleaves heparan sulfate proteoglycans, a major matrix component of blood vessels. Additionally various lysosome associated and secreted ribonucleases were identified, such as RNASET2 that promote mRNA and non-coding RNA degradation.

We hypothesized that the aggressive pathological features of HGG oligodendroglioma and GBM are due to the interactions of diverse cell types, including tumor glial cells and tumor phagocytic cells, such as macrophages or other immune cells with blood vessels resident in the brain. We have identified a transmembrane protein, TMEM230, that recapitulates pathological blood vessel formation and vascular mimicry, both processes having destructive tissue remodeling capacity. Additionally, we observed that secreted factors from U87-MG glial cells promote cell sprouting and blood-vessel-like formation in endothelial cells. Our results support that TMEM230 may be a regulator in the formation of the invasive and infiltrating behavior observed in diverse and heterogenous cell types in HGG oligodendroglioma and GBM. Transcriptomic analysis of patients with gliomas revealed that TMEM230 as membrane protein may regulate genes associated with the motor-protein-dependent Golgi complex and the endoplasmic reticulum intracellular trafficking and secretion of factors promoting angiogenesis and microchannel-generating metalloproteinases. Microchannel formation is a property of phagocytic cells and in the context of glial tumor formation is also referred to as tumor microtubules [22,23,24].

## 4. Materials and Methods

### 4.1. Patient Data Collection

mRNAseq datasets and corresponding patient clinical data were obtained from The Cancer Genome Atlas (TCGA) (Cancer Genome Atlas Research Network), analyzed using R package TCGA2STAT V 5.2.3 [33] and normalized with RSEM [34]. Patients included for analysis are 172 brain samples from patients with high-grade (G3, G4) glioblastoma multiforme (GBM), 198 oligodendroglioma samples, and 197 astrocytoma samples. Grades of tumors were defined according to the American Joint Committee on Cancer, AJCC.

### 4.2. Patient RNA-Seq Gene Expression Analysis

Differential gene expression analysis was performed using DESEQ2 with a *p*-value cut-off < 0.05 and an absolute log2 fold change cut-off > 0.58. Functional enrichment analysis was performed using DAVID (6.8) [35]. Only terms with a corrected *p*-value (Benjamini) < 0.05 were considered [36]. and gene expression analysis was performed using the DESEQ2 R package (version 1.30.1) [37].

### 4.3. Cloning of Lentiviral-System-Based Construct for Inhibiting TMEM230 Protein Expression

The shTMEM230 sequence (for downregulation of endogenous TMEM230) was cloned into pcDNATM6.2-GW/EmGFP using the BLOCK-iTTM Pol II miR RNAi Expression Vector Kit with EmGFP (K493600, Thermo Fisher Scientific, Waltham, MA, USA) following the manufacturer’s instructions. The following sequences were annealed to generate double-stranded oligonucleotides: TOP:5′-TGCTGTGTAGGTTCACTTAACATCTTgttttggccact gactgacAAGATGTTGTGAACCTACA-3′ and BOTTOM:5′-cctgTGTAGGTTCACAACATCTTgtcagtcagtggccaaaacAAGATGTTAAGTGAACCTACAC-3′. Capital letters represent the sense and anti-sense sequences of the small hairpin RNA to be expressed for targeting the endogenous TMEM230 transcript. Lowercase letters are the sequence forming the loop of the hairpin structure. The expression cassette of the resulting plasmid and the control vector provided in the kit (pcDNATM6.2- GW/EmGFP-miR-neg Control/shSCR) were amplified by PCR using the following primers: FW 50-GGCATGGACGAGCTGTACAA-3′ and RVNotI 5′-GTGCGGCCGCATCTGGGCCATTT-3′ (which added a NotI restriction site). The PCR products were cloned into the destination lentiviral vector pCDH-CMV-MCS-EF1-copGFP (CD511B1, System Bioscience, Palo Alto, CA, USA) between BamHI and NotI restriction sites. Lentivirus particles were produced in HEK293T cells by transfecting pCDH or pLENTI vectors together with psPAX2 and pMD2.G (gift from Didier Trono, Addgene plasmids #12260 and #12259) as helper vectors for 2nd-generation viral packaging (with a ratio 4:3:1, respectively) using the LipofectamineTM 2000 Transfection Reagent (11668027, Thermo Fisher Scientific Waltham, MA, USA) following manufacturer’s instructions. Cell culture supernatants containing the lentiviral particles were harvested after 48 and 72 h, concentrated by ultracentrifugation at 120,000 rcf for 3 h, and stored at −80 °C for later use.

### 4.4. Generation and Cloning of the Endogenous TMEM230 Variant 2 (ISOFORM 2) Transcript

The TMEM230 coding sequence was amplified from cDNA obtained from U87 cDNA using primers T230infFw: 5′-gagctagcgaattcgaaTGTTATGATGCCGTCCCGTA-3′ T230infRv and 5′-atccgatttaaattcgaaCTATGGGGTGGGTGCTA-3′. Capital letters represent the nucleotide sequence that anneals with the endogenous TMEM230 transcript. Lowercase letters are the docking sequences of the vector. The destination plasmid pCDHCMV-MCS-EF1-copGFP (SBI CD511B-1) was linearized using the BstBI restriction enzyme. Plasmid insert cloning was completed using In-fusion Cloning Plus (638920, Clontech TAKARA Bio, San Jose, CA, USA) following manufacturer’s instructions. The U87 cDNA sequence was compared to the wild-type sequence on non-malignant human patient cells to confirm that U87 cells did not contain a mutated or aberrant sequence of TMEM230.

### 4.5. Adherent Cell Cultures

The human brain glioblastoma U87-MG cell line was obtained from the ATTC and maintained in DMEM (ECB7501L Euroclone, Pero Mi, Italy) supplemented with 10% fetal bovine serum (FBS, F7524, Sigma, St. Louis, MO, USA), 1% glutamine (BE17-605E, Cambrex, Paullo Mi, Italy), and 1% penicillin/streptomycin (15140-122, Life Technology, Carlsbad, CA, USA) in a humidified atmosphere of 5% CO_2_ at 37 °C. Cells were cultured to an 80% level of confluence. Transduction was performed on adherent cells using lentiviral vectors (shSCR-GFP, used as control, and shTMEM230-GFP for downregulating endogenous TMEM230). Human umbilical vein endothelial cells (HUVECs) were grown in EGM2 medium (CC-3162 Euroclone) and Ham’s F12/DMEM-Glutamax (21765-029:31966-021, Life Technologies, Carlsbad, CA, USA) at a ratio of 1:1 supplemented with additional factors: heparin (CC-4396A), hydrocortisone (CC-4112A), epidermal growth factor (CC-4317A), human basic fibroblast growth factor (CC-4113A), vascular endothelial growth factor (CC-4114A), ascorbic acid (CC-4116A), FBS (CC-4101A), gentamicin (CC-4381A), and R3 insulin-like growth factor (R3IGF1, CC-4115A) all from Euroclone. Human umbilical vein endothelial cells were cultured in a humidified atmosphere of 5% CO_2_ at 37 °C to an 80% level of confluence and medium was replaced twice a week. The human brain astrocytoma cell line 1321-N1 was kindly provided and validated by Prof. Valerio Magnaghi from the University of Milan. Cells were maintained in DMEM supplemented with 10% FBS, 1% glutamine, and 1% P/S in a humidified atmosphere of 5% CO_2_ at 37 °C and cultured to 80% level of confluence. As for U87-MG cells, transduction was performed on adherent cells using lentiviral vectors (shSCR-eGFP, used as control, and shTMEM230-eGFP for downregulating endogenous TMEM230).

### 4.6. Adherent Co-Cultures of Human Umbilical Vein Endothelial Cells and U87-MG Cells

Briefly, 20,000 HUVECs were plated and cultured to confluency. shSCR or shTMEM230 + eGFP-transduced U87-MG cells at low concentration were then added on top of the confluent HUVECs. U87 cells were distinguished from HUVECs because of their green fluorescence. A combination of U87 and HUVEC (EGM2) media at a ratio of 1:1 was used as described in [5]. Cell migration and infiltration of U87 into HUVECs were monitored for 10 days. Half of media was replaced with fresh media every 3 days. All assays were performed in 3 replicates simultaneously and in 3 independent experiments.

### 4.7. Microchannel Formation Assay

shSCR or shTMEM230 + eGFP lentivirus-transduced U87-MG (20,000 cells) were cultured on top of growth-factor-reduced Matrigel (356231, BD Biosciences, Milan, Italy) bed in 48-well plates (677180, Greiner, Twin-Helix, Rho Mi, Italy) using U87 (tumor) or HUVEC (vascular) medium as described in [5]. All assays were performed in 3 replicates simultaneously in 3 independent experiments.

### 4.8. 1321-N1 Microchannel Formation Assay

shSCR or shTMEM230 lentivirus-transduced 1321-N1 astrocytomas (20,000 cells) were cultured on top of growth-factor-reduced Matrigel (BD Biosciences, 356231) bed in 48-well plates (677180, Greiner, Twin-Helix) using 1321-N1 medium, as described for U87 cells in [5]. 

### 4.9. Angiogenesis Assays

For tubulogenesis assay, 20,000 HUVECs were plated on top of growth-factor-reduced Matrigel (356231, BD Biosciences, Milan, Italy) in 48-well plates (677180, Greiner, Twin-Helix) for 24 h. Tubule-forming media were conditioned media obtained from U87shSCR and U87shTMEM230+eGFP. All assays were performed in 3 replicates simultaneously in 3 independent experiments. For spheroid outgrowth assay, 16,000 transduced HUVECs were suspended in 100 mL of 20% HUVEC medium containing 2% methylcellulose solution (M7027, Sigma) in 96-well plates as described in [5]. Spheroids were collected the day after and embedded in 60% methylcellulose containing 40% FBS and Collagen R (SE4725401, SERVA, Euroclone, Pero MI, Italy) at a ratio 1:1 and then layered onto a solidified bed of rat collagen in 96-well plates, as described in [5]. After the methylcellulose/collagen mixture solidified, medium with or without angiogenic-promoting factor and 20 ng/mL VEGF (V7259, Merck, Darmstadt, Germany, V7259). Spheroids formed within 24 h from control cells with VEGF. Four distinct culture conditions were examined: eGFP HUVEC-transduced control cells with or without VEGF; shSCR+eGFP-transduced HUVEC control cells with or without VEGF; TMEM230mRNA+eGFP-transduced HUVECs; and shTMEM230+eGFP-transduced HUVECs. Spheroids from all experimental conditions were compared to spheroids generated from control cells cultured in fresh EGM2. All assays were performed in 3 replicates simultaneously in 3 independent experiments.

### 4.10. Hypoxia Assay

Culture conditions were identical to normoxia except that a hypoxic environment was generated using nitrogen for displaying oxygen.

### 4.11. Western Blotting

HUVECs were lysed on ice using Laemmli buffer, as previously described [5]. Briefly, 20 µg of total protein for each sample was mixed with a 6x loading dye buffer and loaded onto 10% SDS denaturing poly-acrylamide gels. After transferring proteins to a PVDF membrane (10600021 Euroclone), the membrane was blocked with 5% fat-dried milk (EMR180001 Euroclone) and incubated with primary polyclonal rabbit anti-TMEM230 (1:2500, 21466-1-AP, Proteintech, Rosemont, IL, USA,) and polyclonal goat anti-Lamin A/C (sc 376248 Santa Cruz Biotechnology, Dallas, TX, USA) used as endogenous control at concentration 1:7500. Donkey anti-rabbit (1:20,000, NA934V, Amersham, Cologno Monzese, Mi, Italy) and sheep anti-mouse (1:10,000, NA931V Amersham) were used as secondary antibodies.

### 4.12. Data Collection and Statistical Analysis

Transcriptomic profiling of genes in different gliomas was performed using public datasets of oligodendroglioma, GBM and astrocytoma from The Cancer Genome Atlas (TCGA) RNA sequencing (RNAseq) database (Cancer Genome Atlas Research Network) based on the expression level of TMEM230. RNA sequencing data and clinical data of 198 oligodendroglioma, 172 GBM, and 197 astrocytoma patient samples were analyzed for TMEM230 expression level. The samples were divided into TMEM230 high and TMEM230 low based on TMEM230 expression level, as described in [5]. Differentially expressed genes associated with increased glioma tumor grade were identified in LGG and HGG patient datasets using *p*-values and log2 FC, as described in each table. Gene ontology and biological pathways were assessed using the False Discovery Rate method (Benjamini) [36].

### 4.13. Microcopy and Imaging

Imaging was performed with an Olympus IX51 fluorescent microscope (Olympus Italia, Segrate MI, Italy) and XM10 Camera (Olympus Italia) and visualized with Cell F imaging software (version 5.1.2640, Tokyo, Japan). Figure 7 was created by Biorender.com scientific image and illustration software (https://www.biorender.com/, accessed on 21 December 2023).

### 4.14. Supplementary Tables (Excel Files)

Appendix A (sheets 1, 2, and 3) shows all genes showing base mean expression, log fold change (log2), and their adjusted *p*-values for patients with oligodendroglioma, GBM, and astrocytoma, respectively. The gene expression fold changes are correlated with a corresponding differential expression of TMEM230, with an adjusted *p*-value of ≤0.05 for TMEM230 (Appendix A).

## 5. Conclusions

Expression analysis performed in this study supports that TMEM230 is necessary for the endomembrane-dependent intracellular trafficking and secretion of microchannel- and angiogenesis-promoting factors. Our results support that the aggressive tumor behaviors of GBM and HGG may be associated with aberrantly high levels of TMEM230. Progress in glioma research would greatly benefit in understanding which pathways regulated by TMEM230 in endothelial and glial cells contribute to de novo formation of defective blood vessels or vascular mimicry in highly vascularized tumors. Some research supports that infiltrating glioma, such as oligodendroglioma and astrocytoma, gradually become more aggressive and thereby increase in tumor grade over time. If aberrant elevated levels of TMEM230 promote aggressive tumor development, TMEM230 may represent a promising target for cancer therapeutics. Aberrantly formed vessels and microchannels contribute to the inability to target therapeutic agents to the tumor mass.

## 6. Patents

Ileana Zucchi and Rolland Reinbold are recipients of EU Patent EP18707150.1, 2022-09-06 and US Patent US11566070B2, granted on 2023-01-31.

## Figures and Tables

**Figure 1 ijms-25-03967-f001:**
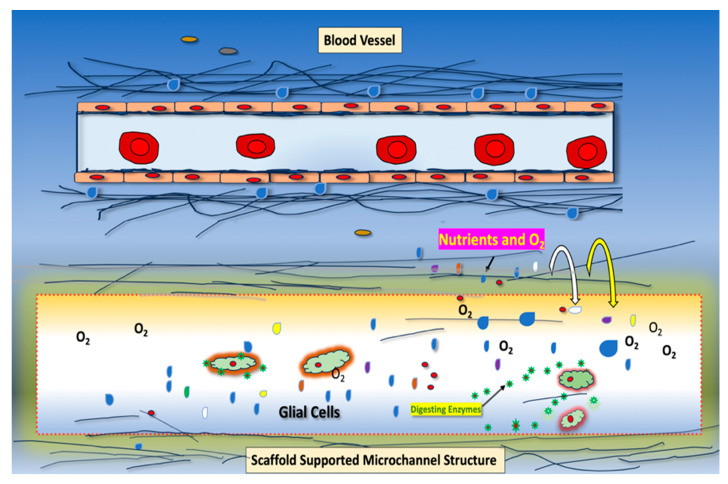
Comparison of blood vessel (**top**) and vascular mimicry promoting microchannel lumen structures (**bottom**) generated by epithelial or glial cells, respectively. Microchannels are generated by phagocytic cells, such as macrophages or glial cells, that secrete enzymes that digest protein and glycan scaffolds and other components of the tissue. Bona fide tubules are lumen containing structures that are supported by a wall of cells that have cell-to-cell or cell-to-substratum contacts. In glial tumor formation, microchannels are often referred to as tumor “microtubules”. In contrast to bona fide cell-lined tubules, tumor microchannels/microtubules are not cell supported structures but are supported instead by scaffolds or extracellular matrix of the tissue in which they are generated. Therefore, scaffold supported microchannels/microtubules consist of luminal 3D “space” that are physically unstable and transient, depending on environmental forces that induce tissue compression.

**Figure 2 ijms-25-03967-f002:**
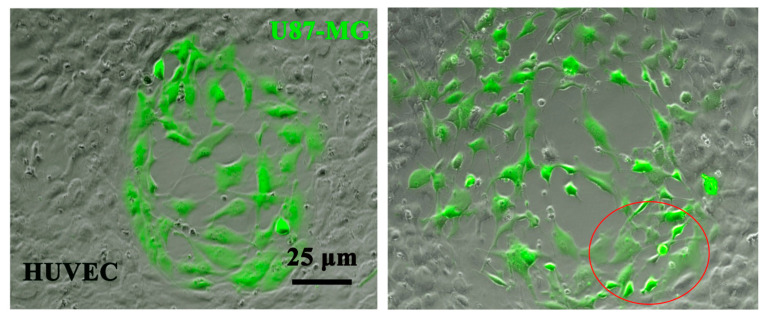
Endogenous TMEM230-promoted U87-MG (shSCR+eGFP control cells) infiltration, disruption, and displacement of human umbilical vein endothelial cells (HUVECs) in co-culture assays. Representative images of U87 shSCR+eGFP-expressing cells plated directly on top of confluent HUVECs at day 9. U87 control cells expressing endogenous TMEM230 infiltrate into the confluent mass of HUVECs (see red circle), a behavior that is associated with the first step of intussusceptive induced blood vessel sprouting, branching, and infiltration of high-grade glioma or glioblastoma multiforme tumors. U87-MG cells in which TMEM230 expression was ablated displayed inability for cell anchorage and endothelial cell contact.

**Figure 3 ijms-25-03967-f003:**
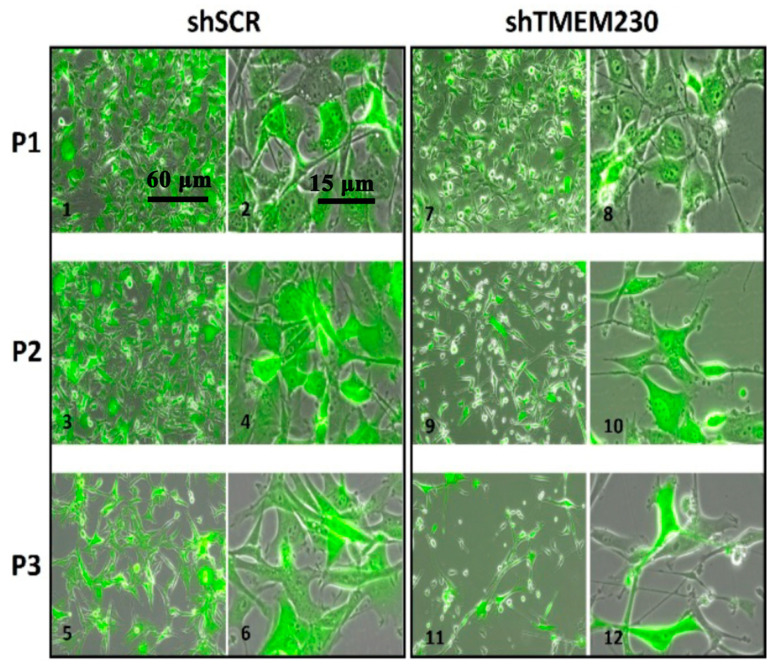
Downregulation of endogenous TMEM230 was sufficient to promote loss of U87-MG substratum adhesion capacity and survival. U87 control (shSCR+eGFP, panels 1–6) and U87 cells in which TMEM230 was constitutively downregulated (shTMEM230+eGFP, panels 7–12) were cultured in 2D conditions. When TMEM230 was downregulated (shTMEM230+eGFP, panels 7–12), U87-MG lost normal cell morphology and cellular cytoplasmic-like invadopodium and projections and detached from the culture plates (panels 7–12), Equal numbers of control cells and cells in which TMEM230 was downregulated were plated. “P” is passage number, where each passage was every 3 days. Panels 1, 3, 5, 7, 9 and 11: lower magnifications, 60 μm; Panels 2, 4, 6, 8, 10 and 12: higher magnifications, 15 μm.

**Figure 4 ijms-25-03967-f004:**
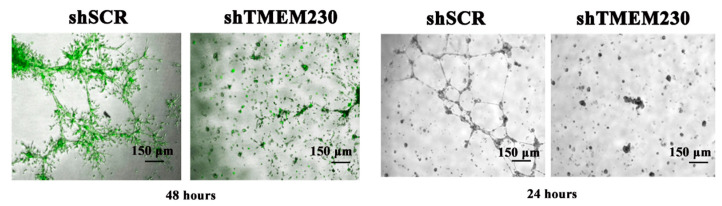
Endogenous expression of TMEM230-promoted U87-MG migration, “tunneling”, and microchannel structure formation, recapitulating extracellular matrix infiltration and vascular mimicry. (Left, 1st panel): representative 3D bodies and lumen structures generated by U87 control cells expressing endogenous TMEM230 and eGFP reporter genes (shSCR+eGFP). Cells cultured in Matrigel at 48 h displayed microchanneling, cell sprouting, collective cell movement, infiltration, and invasion capacity. (Left, 2nd panel): U87 cells in which endogenous TMEM230 was downregulated (shTMEM230+eGFP) did not generate 3D bodies and microchannel structures of significant size, in agreement with TMEM230 being required for cell growth, migration, and survival. (Left, 3rd panel): HUVECs treated with conditioned media obtained from U87-MG cells expressing endogenous TMEM230 promoted angiogenic and cell infiltration behavior. HUVECs cultured in Matrigel with conditioned media obtained from 3-day cultures of U87 in which TMEM230 was downregulated (shTMEM230+eGFP) did not generate microchannel structures (4th panel).

**Figure 5 ijms-25-03967-f005:**
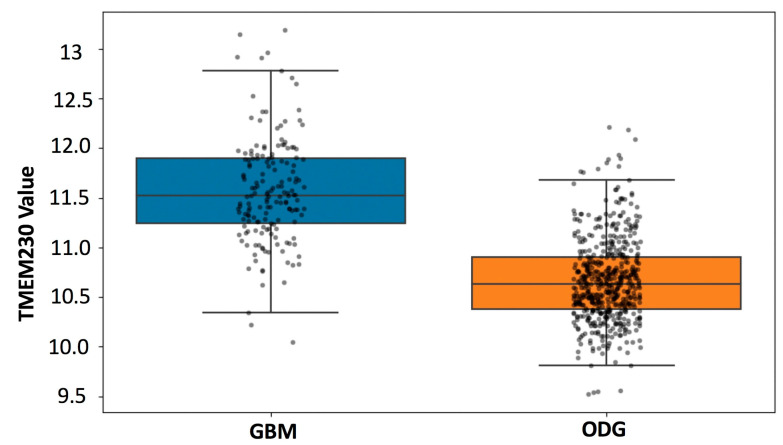
Expression level of TMEM230 in oligodendroglioma and GBM. Glioblastoma multiforme tumors showed significantly elevated level of TMEM230 mRNA compared to oligodendroglioma (unpaired *t*-test *p* < 0.0001).

**Figure 6 ijms-25-03967-f006:**
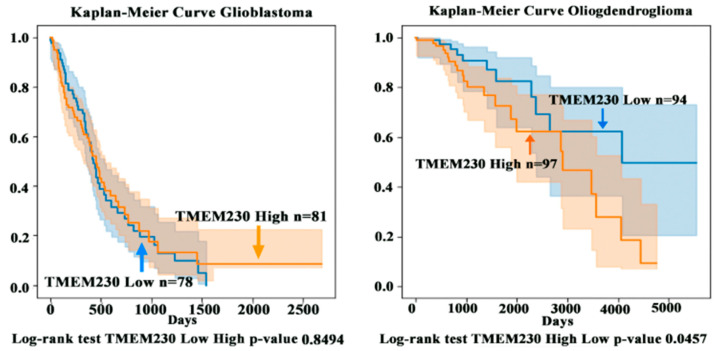
Expression of TMEM230 in low-grade (LGG) and high-grade (HGG) gliomas analyzed from The Cancer Genome Atlas. Kaplan–Meier survival analysis correlated poor prognosis with high TMEM230 expression level. The vertical axis is the probability of the patient surviving, and the horizontal axis is the number of days. The values are expressed in log2 of the number of normalized reads.

**Figure 7 ijms-25-03967-f007:**
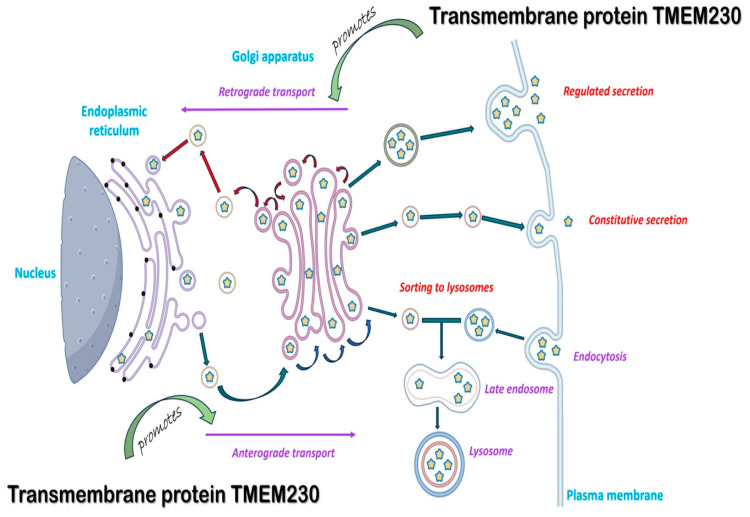
Diagram describing the endomembrane cellular functions of TMEM230 in endoplasmic reticulum and Golgi-dependent intracellular trafficking and secretion of scaffold-digesting enzymes and glycoprotein processing in angiogenesis. The direction of red and dark green arrows corresponds to the retrograde and anterograde transport directions, respectively. Yellow stars correspond to newly synthetized and transported proteins through the endoplasmic reticulum and Golgi complex. Part of the figure was created by Biorender.com (see Appendix A).

**Figure 8 ijms-25-03967-f008:**
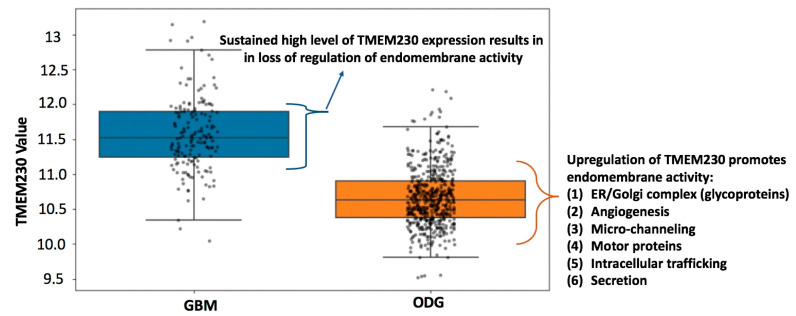
Summary of candidate cellular functions in infiltrating HGG oligodendroglioma and GBM associated with high TMEM230 levels.

**Figure 9 ijms-25-03967-f009:**
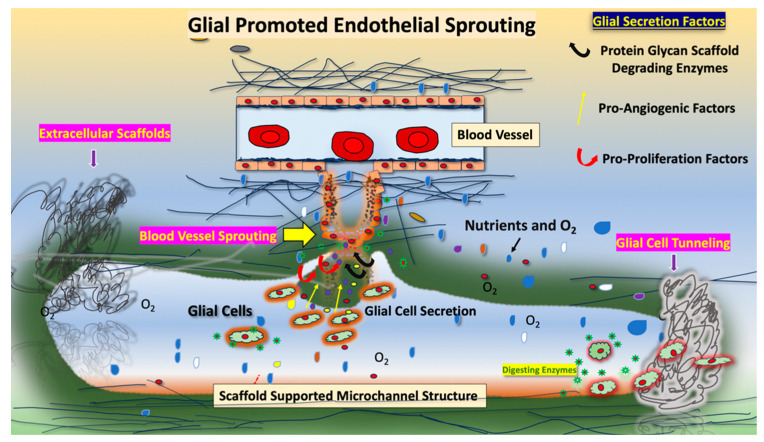
Summary of candidate cellular functions associated with upregulation of TMEM230 in HGG oligodendroglioma and GBM. Collectively, our results support that TMEM230 promotes dynamic interaction of glial secreting cells with endothelial cells, resulting in blood vessel remodeling and microchannel formation. High-grade gliomas (HGG) and glioblastoma multiforme (GBM) are characterized by a heterogeneous and aggressive population of tissue-infiltrating phagocytic cells that promote both destructive tissue remodeling and aberrant vascularization of the brain by secretion of metalloproteinases or ribonucleases, such as RNASET2. Formation of defective and permeable blood vessels and microchannels and destructive tissue remodeling prevent vascular delivery of pharmacological agents to tumor cells and are a significant reason why therapeutic chemotherapy and immunotherapy intervention are primarily ineffective. Yellow arrows, glial and macrophage cells secrete pro-angiogenic factors that induce blood vessel remodeling, sprouting, and branching. Red arrows, glial and macrophage cells secrete factors that promote endothelial cell proliferation and cell infiltration. Black arrows, glial and macrophage cells secrete scaffold degrading enzymes such as heparanase that cleaves heparan sulfate proteoglycans, a major matrix component of blood vessels.

**Figure 10 ijms-25-03967-f010:**
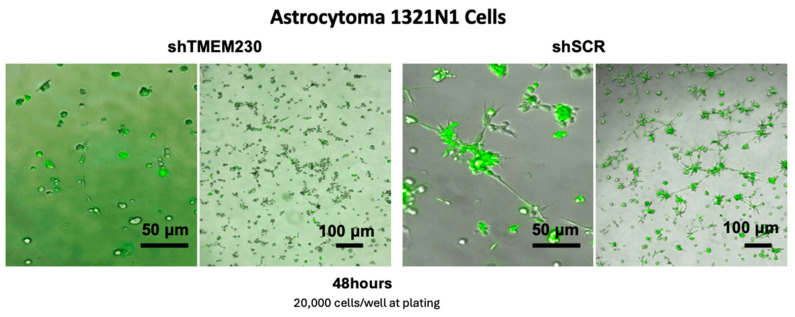
Endogenous expression of TMEM230 promoted 1321N1 astrocytoma migration, “tunneling”, and microchannel structure formation, recapitulating extracellular matrix infiltration and vascular mimicry. (Left, 1st and 2nd panels): 1321N1 cells in which endogenous TMEM230 was downregulated (shTMEM230+eGFP) did not generate 3D bodies and microchannel structures of significant size, in agreement with TMEM230 being required for cell growth, migration, and survival. (3rd and 4th panels): representative 3D bodies and lumen structures generated by 1321N1 control cells expressing endogenous TMEM230 and eGFP reporter genes (shSCR+eGFP). Cells cultured in Matrigel at 48 h.

**Figure 11 ijms-25-03967-f011:**
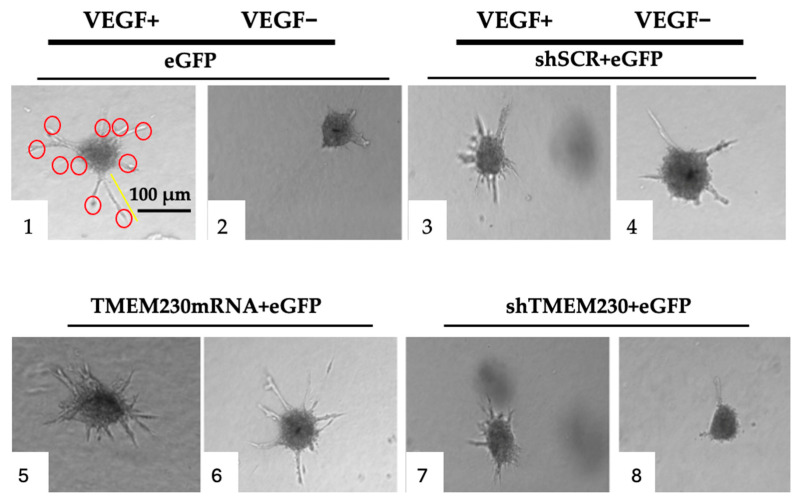
3D cultures were treated with VEGF ligand (20 ng/mL), and TMEM230 expression was modulated in HUVECs. Panels: VEGF ligand (20 ng/mL) was added or not added to control cells expressing eGFP, (1 and 2, respectively); VEGF ligand (20 ng/mL) was added or not added to cells expressing shSCR+eGFP (3 and 4, respectively); VEGF ligand (20 ng/mL) was added or not added to cells in which TMEM230 mRNA was upregulated (TMEM230 mRNA+eGFP) (5 and 6, respectively); and VEGF ligand (20 ng/mL) was added or not added to cells in which TMEM230 was downregulated (shTMEM230+eGFP) (7 and 8, respectively). Experimental assays were performed 3 times, with 3 replicates for each assay. VEGF ligand concentration was 20 ng/mL. Red circles indicate individual branching sprouts. Yellow bar indicates length of a representative sprout.

**Figure 12 ijms-25-03967-f012:**
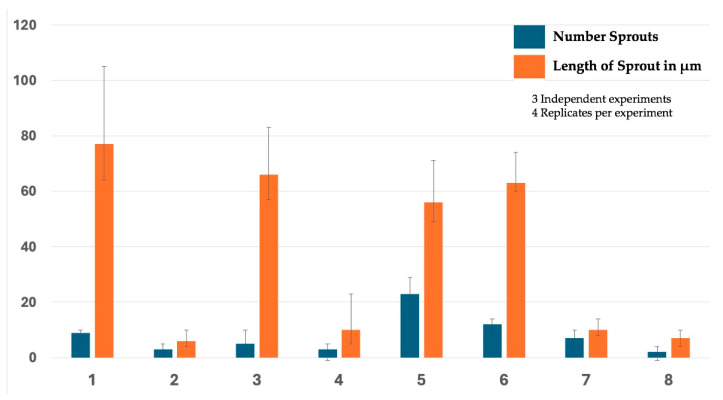
Number and lengths of sprouts quantified in cultures treated with VEGF ligand (20 ng/mL) and in which TMEM230 expression was modulated in HUVECs, corresponding to Figure 11. Panels: VEGF ligand (20 ng/mL) was added or not added to control cells expressing eGFP (1 and 2, respectively); VEGF ligand (20 ng/mL) was added or not added to cells expressing shSCR+eGFP (3 and 4, respectively); VEGF ligand (20 ng/mL) was added or not added to cells in which TMEM230. mRNA was upregulated (TMEM230 mRNA+eGFP) (5 and 6, respectively); and VEGF ligand (20 ng/mL) was added or not added to cells in which TMEM230 was downregulated (shTMEM230+eGFP) (7 and 8, respectively).

**Figure 13 ijms-25-03967-f013:**
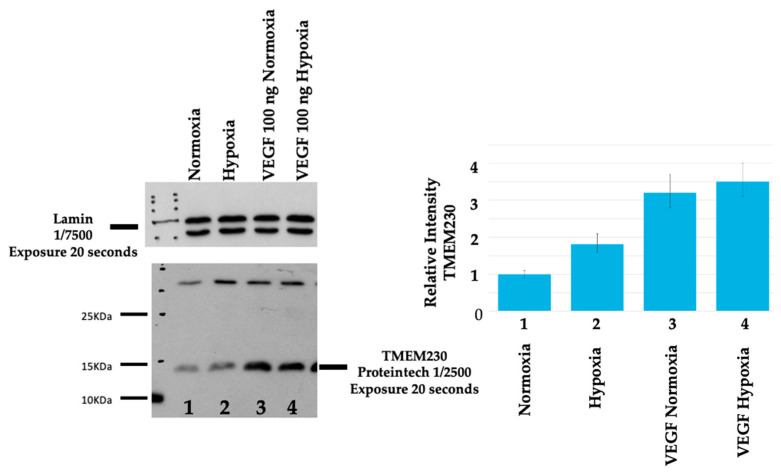
TMEM230 expression was upregulated by hypoxia signaling and independently with VEGF ligand treatment of HUVECs. Upregulation of TMEM230 was observed in cells cultured in hypoxia (lane 2 compared to lane 1) or with cells treated with VEGF ligand (100 ng/mL) in normoxia (lane 3 compared to lane 1). TMEM230 upregulation was less with hypoxia than with treatment with VEGF ligand (lane 3 compared to lane 4 and lane 2 compared to lane 3). Relative intensity was determined by comparing TMEM230 with lamin AC protein expression in Western blot analysis, as described [29].

## Data Availability

This study did not generate gene datasets. No publicly archived datasets were analyzed or generated during the study.

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
