# Peer review of "Transmembrane Protein TMEM230, Regulator of Glial Cell Vascular Mimicry and Endothelial Cell Angiogenesis in High-Grade Heterogeneous Infiltrating Gliomas and Glioblastoma"

_ijms, 2024, doi:10.3390/ijms25073967_

Round 1

Reviewer 1 Report

Comments and Suggestions for Authors

Comments to authors-2024-02-12

Submission ID; ijms-2886023

Thank you very much to give me an opportunity to review this manuscript.

The paper described that the effect of ransmembrane protein TMEM230 on vascular mimicry via acceleration of endothelial and glioma cell invasion. In addition, authors demonstrated that TMEM230 was a prognostic biomarker for oligodendrogliomas, although there was no significance in patients with GBM.

The paper was well-written. I have several requests and questions to authors.

#1. Have authors examined another glioma cell lines in addition to U87-MG? I am wondering whether or not molecular profiles of glioma cells such as p53, MGMT, and IDH-1 status might have an impact on inducing vascular mimicry under TMEM230 exposure.

#2. I would request authors to present the in vivo data.

Does shTMEM230 inhibit the growth of the tumor and angiogenesis in vivo model such subcutaneous or intracranial implanted animal models? Histopathological evaluation of the tumor implanted intracranially by microvessel counting and invasiveness as well as vascular mimicry or vascular cooption are of my great interest.

#3. How does the level of TMEM230 expression alter when the tumors treated by angiogenesis inhibitors such as anti-VEGF antibody or blocking molecules regarding tumor invasion such as MMP or TIMP-1?

#4. Does the expression of TMEM230 influenced by hypoxia or oxygenation? In my opinion, it would be interest and important to investigate whether or not TMEM230 might be active under hypoxic tumor microenvironment. References should be added if previous papers mentioned regarding this.

Author Response

Dear Reviewer we have included our responses to your suggestions, advice and requests, as an attached PDF. The PDF included new analysis and cell functional work that we hope will satisfy your recommendations.

As the new data is extensive we ask the reviewer which data if any in our response should be included in the revised manuscript.

We thank you again for you time in helping us improve the manuscript.

Reviewer 2 Report

Comments and Suggestions for Authors

Researchers, using patient data and cell experiments, explored how TMEM230 influences genes linked to cell infiltration and blood vessel formation in aggressive brain tumors such as high grade gliomas (HGG) and glioblastoma multiforme (GBM).

The study provides strong evidence supporting their claims and is well-structured and clear.

However, minor grammatical errors could be addressed for optimal clarity.

Author Response

Response to Reviewer 2.

Comments and Suggestions for Authors

Researchers, using patient data and cell experiments, explored how TMEM230 influences genes linked to cell infiltration and blood vessel formation in aggressive brain tumors such as high-grade gliomas (HGG) and glioblastoma multiforme (GBM). The study provides strong evidence supporting their claims and is well-structured and clear. However, minor grammatical errors could be addressed for optimal clarity.

We are delighted that the reviewer found our research of interest and has provided positive feedback. The suggestions provided has helped us improve manuscript

We have revised the manuscript at key locations to address the reviewer’s concerns where grammatical errors have been found.

We thank the reviewer again for their time.

We thank the reviewer for helping us to significantly improve this research study.

Reviewer 3 Report

Comments and Suggestions for Authors

Cocola et al report on pathways regulating the expression of the trans-membrane protein TMEM230 in gliomas. A good number of results has been obtained in the U87 model of glioma cells, rather criticised for its low resemblance with real-life glioblastoma (Lenting et al., 2017; Haddad et al., 2021).

The authors show that decreased TMEM230 expression in U87 cells affects vascular mimicry and cell survival (fig. 2), loss of cell adhesion (fig. 3) and loss of migration and ‘tunneling’ (fig. 4). Authors also show that TMEM 230 is more expressed in glioblastoma (GBM) than oligodendrogliomas (ODG) (fig. 5).

There is a claim for association between high TMEM 230 and shorter survival in GBM, not substantiated by the KM analysis of fig. 6 (left panel), while the difference could be significant in ODG (fig.6, right panel).

Fig. 7 and 8  (these are actually Tables) show biological pathways associated with high TMEM230 expression: legends however mention association with glioma grade rather than TMEM230 expression.  A ranking of the pathways, based on statistical significance, is missing so that their association may look somehow arbitrary. Similar considerations can be repeated  for fig. 10-12. Fig.12 in particular, is not explained in the text.

Essentially all these data had been already provided by Cocola et al in 2012 in their Frontiers paper quoted as ref. 1. In some cases figures look very similar.(eg fig. 4 here and fig.5 in the Frontiers paper)

Haddad, A.F. et al. (2021) ‘Mouse models of glioblastoma for the evaluation of novel therapeutic strategies’, Neuro-oncology advances, 3(1), p. vdab100.

Lenting, K. et al. (2017) ‘Glioma: experimental models and reality’, Acta neuropathologica, 133(2), pp. 263–282.

Comments on the Quality of English Language

To be checked in depth

Author Response

We thank the reviewer for their helpful comments, requests and suggestions. We have responded and provided new analysis and experimental results, included as a PDF file. 

We kindly ask the reviewer to help us determine which of the new analysis and experimental cell work should be included in the revised manuscript.

We thank the reviewer again for the very helpful comments and suggestions, that will help improve the manuscript.
